# ReSMAP: Web Server for Predicting Residue-Specific Membrane-Association Propensities of Intrinsically Disordered Proteins

**DOI:** 10.3390/membranes12080773

**Published:** 2022-08-11

**Authors:** Sanbo Qin, Alan Hicks, Souvik Dey, Ramesh Prasad, Huan-Xiang Zhou

**Affiliations:** 1Department of Chemistry, University of Illinois at Chicago, Chicago, IL 60607, USA; 2Department of Physics, University of Illinois at Chicago, Chicago, IL 60607, USA

**Keywords:** membrane binding, intrinsically disordered proteins, membrane-association propensity, amphipathic helix, intrinsically disordered regions

## Abstract

The functional processes of many proteins involve the association of their intrinsically disordered regions (IDRs) with acidic membranes. We have identified the membrane-association characteristics of IDRs using extensive molecular dynamics (MD) simulations and validated them with NMR spectroscopy. These studies have led to not only deep insight into functional mechanisms of IDRs but also to intimate knowledge regarding the sequence determinants of membrane-association propensities. Here we turned this knowledge into a web server called ReSMAP, for predicting the residue-specific membrane-association propensities from IDR sequences. The membrane-association propensities are calculated from a sequence-based partition function, trained on the MD simulation results of seven IDRs. Robustness of the prediction is demonstrated by leaving one IDR out of the training set. We anticipate there will be many applications for the ReSMAP web server, including rapid screening of IDR sequences for membrane association.

## 1. Introduction

The functional mechanisms of many proteins involve the association of their intrinsically disordered regions (IDRs) with acidic membranes [1]. For example, the Wiskott–Aldrich Syndrome protein (WASP) and its neuronal homologue (N-WASP) are autoinhibited until activated in part by the binding of a disordered basic region with the acidic lipid phosphatidylinositol (4,5)-bisphosphate (PIP_2_) in the plasma membrane, leading to the release of *C*-terminal domains and the initiation of actin polymerization [2]. By attaching its basic domain to the plasma membrane and sequestering PIP_2_, the disordered protein myristoylated alanine-rich *C*-kinase substrate maintains a PIP_2_ reservoir, which can be released by binding with calmodulin [3]. The disordered *C*-terminal domains of the tetrameric ligand-gated ion channel protein NMDA receptor modulate its gating properties [4], likely through membrane association of the membrane-proximal regions. A number of membrane proteins that make up the cell division machinery of *Mycobacterium tuberculosis* have disordered cytoplasmic regions, whose membrane association mediates protein–protein interactions [5], particularly through the reduction in dimensionality. Protein–protein colocalization and interactions mediated by membrane association have also been implicated for the disordered intracellular domains of the prolactin receptor, the growth hormone receptor [6], and the T-cell receptor [7], as well as for the membrane-proximal domain of the sheddase ADAM17 [8]. SepF, a water-soluble protein in *M. tuberculosis* and other bacteria, tethers its *N*-terminal amphipathic helix to the inner membrane, allowing it to act as a membrane anchor for the Z-ring at the start of the cell division process [9]. Membrane targeting of the Src family of kinases is achieved in part by basic residues in the disordered *N*-terminal region [10,11]. Both synaptobrevin-2 and α-synuclein have been suggested to promote membrane fusion by associating with membranes [12,13]. Knowledge of the residue-specific membrane association of these and many other IDRs (or intrinsically disordered proteins) will provide deep mechanistic insight into their functional processes.

Characterizing the residue-specific membrane association of IDRs presets significant challenges. NMR spectroscopy can provide the most detailed information, as done for a few IDRs [5,6,11,12,13]. Molecular dynamics (MD) simulations have recently become accurate for modeling the membrane association of IDRs but, even with GPU acceleration, months of simulation time may be required to cover their vast conformational space [5]. A fast method, such as one based on IDR sequences, is highly desirable. For association with acidic membranes, the importance of basic residues has been a universal observation, whereas the roles of aromatic and hydrophobic residues seem to be context-dependent [1,3].

Recently a sequence-based method has been developed for a related problem, i.e., for predicting the propensities of IDRs binding to nanoparticles [14]. The nanoparticle-binding propensities are predicted from a partition function that is determined by the IDR sequence. Here we adapt this method into a predictor called ReSMAP, for residue-specific membrane-association propensities.

## 2. Computational Methods

The training data for ReSMAP were obtained from molecular dynamics (MD) simulations of seven intrinsically disordered regions (IDRs): *N*-terminal regions of ChiZ, FtsQ, and SepF, which are components of the cell division machinery of *M. tuberculosis*; the membrane-proximal regions in the *C*-terminal domains of the GluN1 and GluN2B subunits of the NMDA receptor; and disordered regions of N-WASP and WASP. In addition, a separate IDR in SepF was simulated and the data were used solely for testing ReSMAP. The two SepF IDRs, spanning residues 1–50 and 66–124, are referred to as SepF1 and SepF2, respectively. The numbers of residues in these IDRs are 64, 99, 50, 105, 123, 85, 82, and 59, respectively (see Appendix A for sequences). The MD simulation protocol was as described by Hicks et al. [5]. The force field combination was AMBER14SB [15] for proteins, TIP4P-D for water [16], and Lipid17 [17] for membranes. The lipid compositions in the simulations of the eight IDRs are listed in Appendix A. The total MD simulation times were 38 μs for ChiZ (among 20 replicate runs), 1 μs each for GluN1, GluN2B, and SepF1 (among 4 replicate runs), 16 μs each for N-WASP, WASP, and FtsQ (among 16 replicate runs), and 2.88 μs for SepF2 (among 8 replicate runs). Except for ChiZ [5], the simulation results were not reported previously and their functional implications will be reported elsewhere.

Previously we measured the level of membrane association by using the contact probability of each residue, i.e., the fraction of MD snapshots in which this residue forms at least one contact, defined with a 3.5 Å cutoff, between its heavy atoms and lipid heavy atoms [5]. Here we introduce another measure, based on *z*_tip_, the mean *z* coordinate of the tip atom of each side chain in a Cartesian coordinate system where the *xy* plane is located on the phosphate plane of the membrane. We convert *z*_tip_ into a contact probability via a smooth function,
(1a)$$C=\frac{1}{1+\exp\left[\left(z_{\mathrm{tip}}-z_{0}\right)/L\left(z_{\mathrm{tip}}\right)\right]}$$
where
(1b)$$L\left(z_{\mathrm{tip}}\right)=L_{1}-\frac{L_{1}-L_{0}}{1+\exp\left[\left(z_{\mathrm{tip}}-L_{m}\right)/L_{w}\right]}$$

Fit to the previous “raw” contact probability yields *z*_0_ = 6.9 Å, *L*_1_ = 3.2 Å, *L*_0_ = 1.2 Å, *L*_m_ = 5.0 Å, and *L*_w_ = 0.5 Å (Appendix A).

Given the universal importance of basic residues in the association with acidic membranes, we first checked whether a toy model, based on the moving average charge calculated over a seven-residue window, had any merit. The charge of each amino acid was +1 (K, R, and the *N*-terminus), −1 (D, E, and the *C*-terminus), or 0 (all other). Even this toy model showed some promise (Appendix A): a linear regression analysis against the MD membrane-contact probabilities yielded coefficients of determination (*R*^2^) in the range of 0.20 to 0.58 for six fully disordered IDRs. Encouraged by the promising sign of the toy model, we sought a more sophisticated method for predicting residue-specific membrane-association propensities by following the basic idea behind the sequence-based partition function of Li et al. [14]. The modifications from the toy model are threefold. First, instead of an abrupt cutoff beyond a sequence distance of 3 residues when considering the effects of neighboring residues, we model the effects with a continuous function that attenuates with increasing sequence distance. Second, instead of an additive term, the effect of each neighboring residue is modeled as a multiplicative factor. Third, the model in theory allows for the amino acids to be divided into more than just three groups (positively charged; negatively charged; and neutral).

Specifically, we assume that the central residue, with index *n*, and all other residues in the sequence each contribute a multiplicative Boltzmann factor to the statistical weight for residue *n*’s membrane association:(2)$$w_{n}=\overset{N}{\underset{i=1}{\prod }}q_{i;\left|i-n\right|}$$
where the residue index *i* runs through the entire sequence (a total of *N* residues), and $q_{i;\left|i-n\right|}$ is the contributing factor of residue *i*. The latter depends on the amino-acid type of residue *i* and the sequence distance |*i* − *n*|:(3)$$q_{i;\left|i-n\right|}=1+\frac{q_{i;0}-1}{1+a_{i}\left|i-n\right|+b_{i}{\left|i-n\right|}^{2}}$$

The dependence on sequence distance is to attenuate the contributing factor as residue *i* moves farther from residue *n* along the sequence. Note that a residue with an amplitude $q_{i;0}>1$ increases $w_{n}$ whereas a residue with $q_{i;0}<1$ decreases $w_{n}$. At present we only distinguish three types of amino acids: positively charged (K, R, and the *N*-terminus), negatively charged (D, E, and the *C*-terminus), and neutral (all other amino acids). We denote the $q_{i;0}$ values of these three types as $q_{+}$, $q_{-}$, and $q_{0}$, respectively. The $a_{i}$ and $b_{i}$ values are the same for all charged residues and are denoted by $a_{\pm }$ and $b_{\pm }$, respectively. For uncharged amino acids, $b_{i}$ = 0 and the common $a_{i}$ value is denoted by $a_{0}$. We inherit distance parameter values from Li et al. [14], with $a_{\pm }$ = 0.0982, $b_{\pm }$ = 0.00305, and $a_{0}$ = 0.521. The amplitude parameters $q_{+}$, $q_{-}$, and $q_{0}$ are optimized against MD data for contact probabilities. The membrane-association propensity is proportional to $w_{n}$:(4)$$P_{n}=c\frac{w_{n}}{w_{\max}}$$
where $w_{\max}$ is the maximum of $w_{n}$ in the entire sequence, and c is a scaling factor (a constant for each protein).

## 3. Results

As reported previously, 13 positively charged R residues in the ChiZ IDR drive its association with acidic membranes [5]. The association is highly dynamic: each moment a different subset of the R residues forms membrane contact (see Figure 1a for a snapshot). The IDRs of ChiZ, GluN1, GluN2B, N-WASP, and WASP remain fully disordered while associating with acidic membranes. Upon optimizing the three parameters $q_{+}$, $q_{-}$, and $q_{0}$, Equation (4) accurately predicts the residue-specific contact probabilities for all these IDRs (Figure 1b,c and Appendix A). The parameter values are $q_{+}$ = 2.43, $q_{-}$ = 0.26, and $q_{0}$ = 0.59 (Appendix A). Only $q_{+}$ is > 1, and so only positively charged residues favor membrane association; both negatively charged and uncharged residues disfavor membrane association, strongly in the former case and weakly in the latter case. The root-mean-square-errors (RMSEs) measured against the MD membrane-contact probabilities are 0.043, 0.035, 0.012, 0.023, and 0.011 for the IDRs in ChiZ, GluN1, GluN2B, N-WASP, and WASP, respectively. These RMSEs are up to 3-fold lower than the counterparts produced by the linear regression equation using the moving average charge (Appendix A). As another measure of accuracy, we display the Equation (4) predictions and the corresponding MD contact probabilities as a scatter plot in Appendix A. The points are all close to the diagonal line *y* = *x*, with a coefficient of determination ($R_{0}^{2}$) at a high value of 0.91. Note that $R_{0}^{2}$ essentially measures the RMSEs against the mean amplitudes of the MD contact probabilities.

To demonstrate the robustness of the predictions, we compare the parameter values when one of the IDRs is left out of the training set. Because two of the IDRs, from N-WASP and WASP, have moderate sequence identity (34%), we left them out together in this exercise. The resulting values are 2.7 ± 0.7 (mean ± standard deviation) for $q_{+}$, 0.28 ± 0.08 for $q_{-}$, and 0.59 ± 0.04 for $q_{0}$, agreeing well with those from the full training set. As one more test, we carried out MD simulations for another fully disordered IDR, SepF2. This IDR is not in the training set but the prediction, with an RMSE of 0.18, also agrees well with the MD results (Appendix A). The $R_{0}^{2}$ value for this test IDR is still high, at 0.73 (Appendix A).

Two other IDRs, FtsQ and SepF1, each have an amphipathic helix (residues 48 to 73 in FtsQ and 1–11 in SepF1) that stably associates with acidic membranes. The first 100 residues of α-synuclein also form amphipathic helices that stably associate with PIP_2_, as characterized recently by NMR spectroscopy [13]. We combined the MD contact data for the FtsQ and SepF1 IDRs and the NMR data for every third residue of α-synuclein to optimize Equation (4). With $q_{+}$ = 2.29, $q_{-}$ = 0.64, and $q_{0}$ = 1.17 (Appendix A), the membrane-contact probabilities of the FtsQ and SepF1 IDRs and the entire α-synuclein sequence are predicted well (Figure 1d and Appendix A). The values of $q_{+}$, $q_{-}$, and $q_{0}$ are essentially unchanged when other one thirds of the α-synuclein NMR data are used for training. The RMSEs are 0.19, 0.27, and 0.17; the $R_{0}^{2}$ value, which as noted above measures the RMSEs against the mean amplitudes of the observed contact probabilities, is a high 0.83 (Appendix A). The prediction accuracy is lower than that for fully disordered IDRs, possibly indicating that the sequence-based partition function, with only three adjustable parameters, is too restrictive for IDRs that have both amphipathic helices and disordered residues (see Discussion). Note that $q_{0}$ is now above 1 (albeit only slightly), likely due to the occurrence of neutral residues in amphipathic helices.

We further tested the foregoing ReSMAP method against residue-specific membrane association data of IDRs that we could find in published NMR studies. Chemical shift differences ($\Delta \delta_{\mathrm{NH}}$) obtained by Haxholm et al. [6] suggested three lipid interaction sites at the *N*-terminus, middle, and *C*-terminus of the intracellular domain (residues G236-H598) of the prolactin receptor. ReSMAP predicts the *N*- and *C*-terminal sites and a middle site that partially overlaps with the NMR data (Figure 2a). In addition, it predicts high membrane-association propensities for the stretch of residues 509–512, with sequence KPKK. The same NMR study also revealed a single lipid interaction site, at the *N*-terminus of the intracellular domain (S270-P620) of the growth hormone receptor. ReSMAP predicts the same site (Figure 2b). Pond et al. [11] reported chemical shift differences for the disordered *N*-terminal region (G2-E79) of the Src kinase Hck, showing a membrane interaction site in residues R18-T40. Our ReSMAP prediction agrees well with this interaction site (Figure 2c). It further predicts high membrane-association propensities for the first three residues (GGR), which might be consistent with the NMR data since resonances of these residues appear to be missing. Missing resonances can result from membrane association. Lakomek et al. [12] studied the membrane association of the soluble portion (residues M1-M96) of the SNARE protein synaptobrevin-2 by NMR. The *C*-terminal 15 or so residues have a high propensity for forming a helix in solution. These residues also show strong binding with liposomes containing 20% DOPS, presumably as an amphipathic helix, as evidenced by missing NMR peaks; the membrane-bound population (as monitored by the ratio of peak intensities in the presence of liposomes and in solution) rapidly decreases toward the *N*-terminus. The ReSMAP prediction shows an identical trend (Figure 2d).

We also tested ReSMAP against experimental data for two other IDRs. Using in-cell FRET, Zhang et al. [7] monitored the binding of the intracellular domain of the T-cell receptor ζ chain to the plasma membrane. Alanine mutations of basic residues in three basic-rich stretches abolished membrane binding, whereas phenylalanine mutations of tyrosines in the so-called intracellular immunoreceptor tyrosine-based activation motifs had no effect. Consistent with these mutation results, ReSMAP predicts high membrane-association propensities for all the three basic-rich stretches and low membrane-association propensities for all the three intracellular immunoreceptor tyrosine-based activation motifs (Figure 2e). Lastly, Sommer et al. [8] identified a basic motif (R625-K626-G627-K628) for mediating interactions of the membrane-proximal domain (F581-E642) of ADAM17 with phosphatidylserine lipids, as indicated by significant chemical shift perturbations in the presence of phosphorylserine. Glycine mutations of the three basic residues abolished binding with phosphatidylserine liposomes. ReSMAP predicts a single high peak for the experimentally identified basic motif (Figure 2f). Together, the experimental data from the six proteins provide mounting evidence in support of the ReSMAP method.

We have implemented ReSMAP as a webserver, accessible at https://pipe.rcc.fsu.edu/ReSMAPidp/. As an illustration of its application, a number of other components of the cell division machinery of *M. tuberculosis*, including FtsB and FtsL, are also likely to associate with acidic membranes via IDRs. We present the predicted membrane-association propensities of these IDRs in Figure 3 and Figure 4.

## 4. Discussion

There is growing recognition of the biological importance of IDR-membrane binding. ReSMAP, by predicting residue-specific membrane association propensities of intrinsically ordered proteins, fulfills an urgent need where little prior work has been conducted. Amphipathic helices have received some attention. In particular, the concept of the hydrophobic moment has been introduced [19] and used [20] to analyze known or predicted amphipathic helices. Some methods have been developed to predict membrane-binding amphipathic helices. One such method is AmphipaSeek (https://npsa-prabi.ibcp.fr/cgi-bin/npsa_automat.pl?page=/NPSA/npsa_amphipaseek.html (accessed on 1 July 2022) [21], which is a support-vector-machine classifier trained on 21 proteins with in-plane membrane anchors centered on one (or more) amphipathic helix. AmphipaSeek predicted only nine residues (V3-A11) of α-synuclein as membrane anchors, even though NMR spectroscopy has shown that the first 100 residues of this protein form amphipathic helices that stably associate with membranes [13]. AmphipaSeek also failed to predict any membrane anchors for either FtsQ or SepF, even though both of them are known to have an amphipathic helix for membrane association (Figure 1d and Appendix A and Ref. [9]). In comparison, our ReSMAP method predicts high membrane-association propensities for all these amphipathic helices (Figure 1d and Appendix A). Another recent method [22], based on a convolutional neural network trained on 121 membrane proteins, failed to predict any amphipathic helix for α-synuclein, FtsQ, and SepF.

The ReSMAP method currently groups amino acids into only three types: positively charged (K, R, and the *N*-terminus), negatively charged (D, E, and the *C*-terminus), and neutral (all other amino acids). This net-charged based grouping is the main limitation of ReSMAP at present and is necessitated by the limited training data. While the latter predicament highlights an important area for future work, it also raises the concern of whether the training set collected here is representative of the association of IDRs with acidic membranes in general. The validation provided by the experimental data gathered from the literature on six more proteins (Figure 2) goes a long way in allaying this concern.

As more MD simulation data for IDR-membrane association become available, it may be sensible to divide up the neutral amino acids (e.g., polar vs. nonpolar), or even separate R and K. Additionally, lipid compositions vary widely in cell membranes, in experimental studies, and even in our MD simulations (Figure 2 caption and Appendix A). In particular, the acidic lipids, including POPS, PIP_2_, and POPG, vary from one study to another. By training on data collected from studies with different lipid compositions, ReSMAP relies on common molecular and biophysical properties of most lipids [23], such as net charge (for acidic lipids), polar headgroups, and hydrophobic tails. It has been suggested that membrane protein structure and function have a high degree of tolerance to the change in lipid composition [24]. Still, membrane-association propensities of an IDR may well change subtly as the lipid composition is varied, and such secondary effects may be important biologically but again will require more MD or experimental data for training. Another subtlety is that IDR association will affect the lipid distribution within a membrane. For example, as illustrated in Figure 1a, acidic lipids (POPG) will cluster around positively charged R residues. ReSMAP does not directly account for such lipid redistribution. Rather, it assumes, for a given residue, an average effect on its own membrane-association propensity and on those of its neighbors.

The prediction accuracy of ReSMAP for IDRs with both amphipathic helices and disordered residues is already good but is still lower than the accuracy for fully disordered IDRs. Amphipathic helices and disordered residues likely have different driving forces for membrane association. Amphipathic helices, by definition, have a charged face and a nonpolar face, and are inserted more deeply into the membrane, such that charged groups interact with lipid headgroups while nonpolar groups interact with lipid acyl tails. In contrast, disordered residues are less buried and mainly interact with lipid headgroups. For IDRs containing amphipathic helices, the ReSMAP optimized parameters are a compromise between the helical and disordered residues. A future development will be to increase the number of parameters and train on a data set with more amphipathic helices.

## Figures and Tables

**Figure 1 membranes-12-00773-f001:**
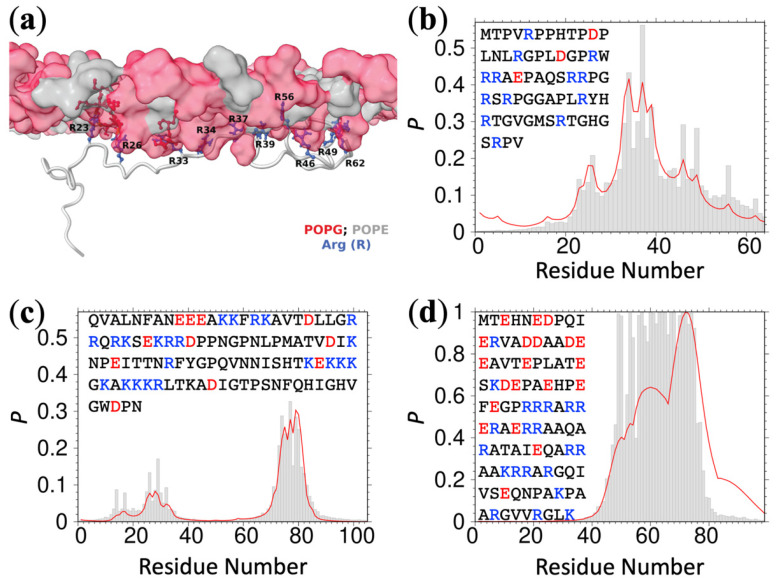
MD data and ReSMAP predictions for membrane-association propensities. (**a**) A snapshot of the ChiZ IDR associated to an acidic membrane, reprinted from Ref. [5]. The lipid headgroups in the leaflet in contact with the IDR are shown as surface; Arg side chains are shown as stick. (**b**–**d**) Comparison of MD membrane-contact probabilities (gray bars) and predicted membrane-association propensities (red curves) for ChiZ, N-WASP, and FtsQ, respectively. The sequence of each IDR is listed, with positively and negatively charged residues colored blue and red, respectively.

**Figure 2 membranes-12-00773-f002:**
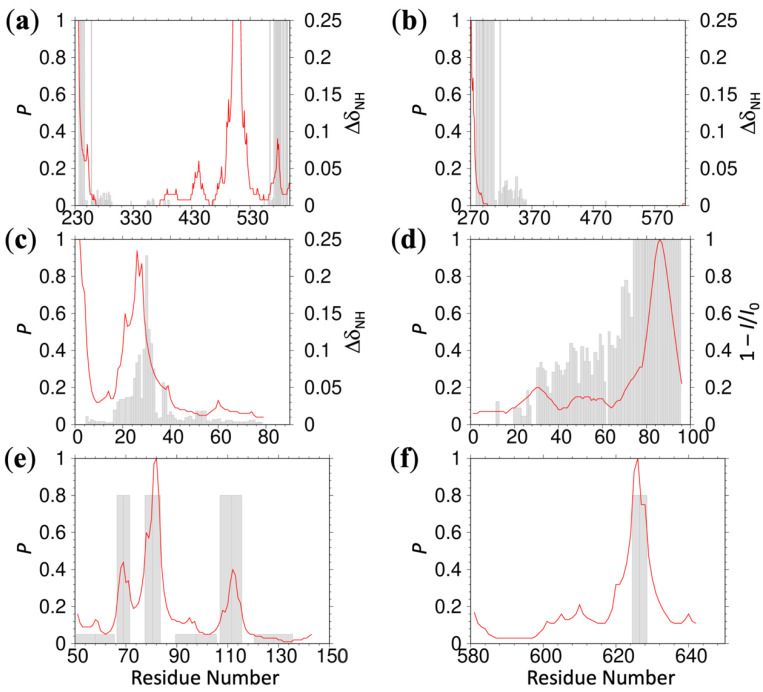
Comparison of ReSMAP-predicted membrane-association propensities (red curves) with experimental data (gray bars). (**a**,**b**) Prolactin receptor and growth-hormone receptor disordered intracellular domains. The experimental data are from Haxholm et al. [6], displaying $\Delta \delta_{\mathrm{NH}}=\sqrt{{\left(\Delta \delta_{H}\right)}^{2}+0.154{\left(\Delta \delta_{N}\right)}^{2}}$, where $\Delta \delta_{N}$ and $\Delta \delta_{H}$ are changes in backbone amide chemical shifts when the IDRs were moved from solution to vesicles formed by POPC/POPS lipids at a 3:1 ratio. $\Delta \delta_{\mathrm{NH}}$ values < 0.005 ppm were assumed to be within experimental error and set to 0; missing resonances were assigned a $\Delta \delta_{\mathrm{NH}}$ value of 0.25 ppm. The protein sequences are from Uniprot (https://www.uniprot.org/ (accessed on 28 July 2022) entries P16471 and P10912, with numbering shortened by 24 and 18 residues (i.e., without the *n*-terminal signal peptides), respectively. (**c**) Hck kinase disordered *N*-terminal region (residues G2-E79). The experimental data are from Pond et al. [11], displaying $\Delta \delta_{\mathrm{NH}}=\sqrt{{\left(\Delta \delta_{H}\right)}^{2}+0.2{\left(\Delta \delta_{N}\right)}^{2}}$ obtained when the IDR was moved from solution to bicelles formed with DMPC:DMPA lipids at a 4:1 ratio. (**d**) Synaptobrevin-2 residues M1-M96. For the experimental data [12], *I*_0_ and *I* were measured, respectively, in solution and in the presence of liposomes formed by DOPC/DOPS/DOPE/Cholesterol at 5:2:2:1. Resonances missing in the presence of liposomes were assigned a zero value for *I*. (**e**) T-cell receptor ζ chain disordered intracellular domain. The protein sequence is from Uniprot entry P24161, with numbering shortened by 21 residues. The tall bars indicate basic-rich stretches where alanine mutations of basic residues abolished plasma membrane binding, as assayed by in-cell FRET [7]. The short bars indicate tyrosine-containing motifs where phenylalanine mutations of tyrosine residues had no effect on plasma membrane binding. (**f**) ADAM17 membrane-proximal domain (residues F581-E642). Bars indicate four residues (R625-K628) that experienced significant chemical shift perturbations in the presence of phosphorylserine [8].

**Figure 3 membranes-12-00773-f003:**
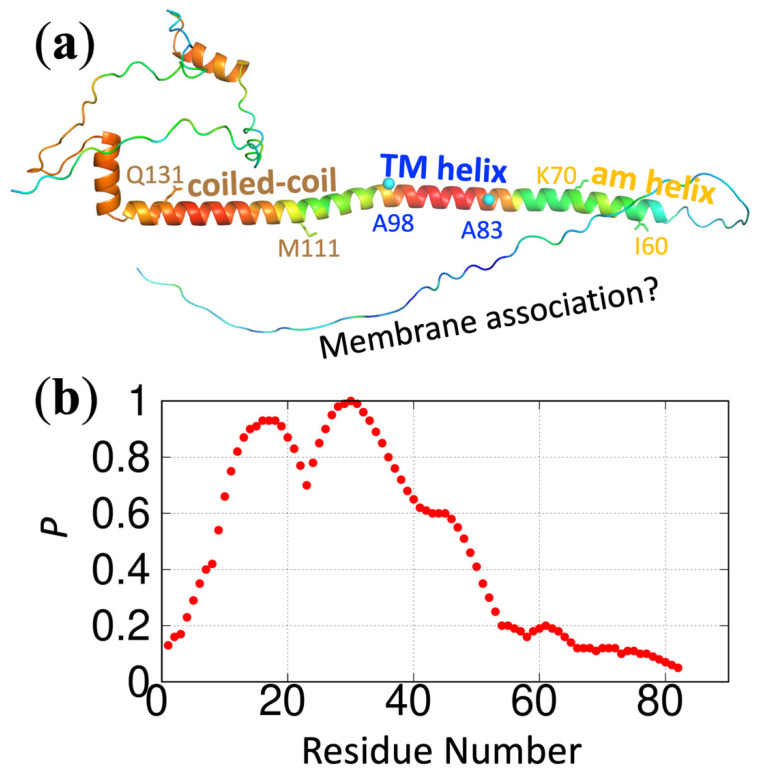
Predicted membrane-association propensities of FtsB. (**a**) Predicted structure by AlphaFold (https://alphafold.ebi.ac.uk/ (accessed on 18 March 2022) [18]. The color spectrum displays confidence levels of prediction, ranging from red for high confidence to blue for low confidence. Three putative helices are indicated with start and end residues: amphipathic (am) helix, residues I60-K70; transmembrane (TM) helix, residues A83-A98; and coiled-coil, M111-Q131. (**b**) Predicted membrane-association propensities using the sequence of the first 82 residues, high for residues 9–48 but only modest for the putative amphipathic helix. The protein sequence is from Uniprot entry P96376.

**Figure 4 membranes-12-00773-f004:**
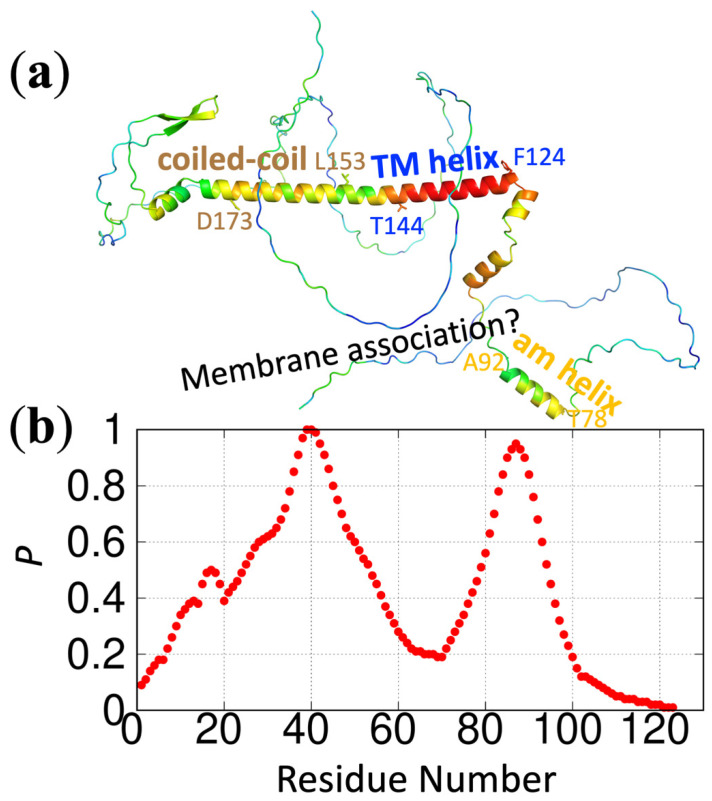
Predicted membrane-association propensities of FtsL. (**a**) Predicted structure by AlphaFold (https://alphafold.ebi.ac.uk/ (accessed on 18 March 2022) [18]. The color spectrum displays confidence levels of prediction, ranging from red for high confidence to blue for low confidence. Three putative helices are indicated with start and end residues: amphipathic (am) helix, residues T78-A92; transmembrane (TM) helix, residues F124-T144; and coiled-coil, L153-D173. (**b**) Predicted membrane-association propensities using the sequence of the first 123 residues, high for residues 25–53 and the putative amphipathic helix. The protein sequence is from Uniprot entry O06213.

## Data Availability

The software developed here is accessible and can be downloaded at https://pipe.rcc.fsu.edu/ReSMAPidp/. The data used for training the method can be requested from the corresponding author.

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
