# Peer review of "ReSMAP: Web Server for Predicting Residue-Specific Membrane-Association Propensities of Intrinsically Disordered Proteins"

_membranes, 2022, doi:10.3390/membranes12080773_

Round 1

Reviewer 1 Report

In the draft, the authors designed a new webserver to rapidly screen membrane association of IDR sequences. However, it has several issues that need to be improved before publishing. 

1. the data set used to train and test the model needs to be improved. In the paper, the training data is from 7 disordered regions, and the test data is based on one disordered region. The number of training and test sequence has to be improved. It is not enough to train, validate and test a model. 

2. How to annotate the ground truth of the data set. It should provide detailed information about where the experimental data is collected from.

3. The method used to validate and independently test the model. The author didn't provide the training and validation results of the model. For example, the AUC, ACC, SEN, F1 or MCC, etc.. and compared the new model ReSMAP with other recent methods, like mentioned in the discussion, the convolutional neural network based method or the AmphipaSeek. In the paper, they provide the case study results, which are not sufficient to evaluate the performance of the model.

Author Response

Response:

  1. We now report test results on six more proteins (lines 170-206; new Figure 2). The training set is sufficient because we have limited our model to only 3 parameters.

  1. We have two sources for the target data, i.e., membrane contact probabilities obtained from molecular dynamics (MD) simulations and inferred from NMR studies. The MD part is explained under Methods (lines 66-91); the NMR part is explained in figure captions (Figures 2 and S5b).

  1. AUC and the other measures are specifically for classification methods whereas we predict propensity, which is a continuous function. So we now report RMSE and R0^2 (lines 137-145, 152-153, and 162-164; new Figure S4). As we noted and also recognized by Reviewer 2 (“there aren’t really any previous predictors out there”), there is no prior methods that we could compare our methods to. The convolutional neural network method and AlphipaSeek do not directly do what our method predicts. These methods predict amphipathic helices, and their performance is very poor when tested on the few proteins that we study and have amphipathic helices (lines 267-276).

Reviewer 2 Report

The manuscript by Qin et al. Describes the development of a web server that predicts interactions between intrinsically disordered regions and negatively charged membrane surfaces. This is an important and general subject and is certainly worth attention. As far as I know, there aren’t really any previous predictors out there. That said, however, I have some reservations about the design and performance of the reported web server. The work is rather preliminary, and the server mainly seems to predict clusters of positive charges, which one could do without training a predictor. Therefore, I am not convinced that the server really advances the state of the field at the present preliminary stage, and I recommend that the work is developed further before publication. Furthermore, the work could be placed better into the scientific literature by including some of the substantial amounts of biophysical studies of membrane interactions of peptides.

Major comments:

Training data and model:

1: The training data set is very small with only 7 proteins/peptides included. This is a clear limitation but is a limitation of the currently available data. However, it raises the question of whether the sample is representative of membrane interactions in general. If possible, a broader data set should be used, but at least the limitations should be discussed clearly.

2: The 7 simulations use different membrane environments, but this is not commented on. Of particular importance is the inclusion of PIP2, which has a strong effect on electrostatic interactions. How can you develop a predictor for interactions without specifying what it should interact with?

3: There are two major modes of interactions of IDRs with membranes. As amphipathic helices or as disordered chains. The sequence signatures of these will not be similar as the amphipathic helices will have a strong periodicity. The training set does not currently take this into account, and it is not clear whether the predictors is meant to predict both types of interactions at once or just one. As I understand the model, it mainly focus on linear sequence distance, which would suggest that the goal is to predict interactions in a random coil conformation, but several of the examples shown in the SI are helical.

4: The model only considers the net charge of residues, and not other factors known to be important such as aromatics and Arg vs Lys. The authors are clearly aware of this (as seen in the discussion). The question is thus whether the server is “good enough” at present, which I am not wholly convinced about. It would be helpful if predictions were done for a broader set of proteins not included in the training data set. At the very least, the limitations of the current algorithm should be spelled out more clearly. Examples with NMR data could include: PMID: 25846210

Presentation and placement in the literature:

5: The introduction is currently very brief, and does not really explain how membrane interactions affect the function of proteins. It would be helpful for the biologically oriented reader if a little more biochemical context could be added. Inspiration and references could be found in this review: PMID: 28601983

6: There has been a substantial amount of research done on interactions of peptides with negatively charged membranes, which is not really discussed in the paper. A good starting point for reading is the works of Stuart MacLaughlin, you may e.g. start with this review: PMID: 16319880. In my understanding, the current state of the art is that clusters of aromatics and positively charged residues will drive interactions with membranes, but that this is heavily dependent on membrane composition. This is similar to what the server predicts (without considering membrane composition or hydrophobicity), so it is not entirely clear that the server really advances the state-of-the-art at this preliminary stage.

7: To make an impact, the paper needs to be read by scientists that do not share the physical background as the authors. Therefore, I would recommend that you try to make the description of the algorithm (section 2) more accessible by describing the design and reasoning behind the algorithm in plain language as well. (You could ask a friendly molecular biologist to read the section and provide

Author Response

Response:

Overall: We have now made a number of improvements. In particular, we show our method is much more accurate than a toy charge model mentioned by the reviewer; we also test our method against experimental data on six more proteins. We have expanded the Introduction to cite many more previous biophysical studies.

1: We appreciate the reviewer’s recognition of the scarcity of the currently available data and also the concern of whether the sample is representative of membrane interactions in general. We now further validate the method by experimental data on six more proteins (including two noted by this reviewer in #4 below). We think this validation largely allays the concern about generality (lines 281-284).

2: We briefly discussed lipid composition in the original submission and now expand this discussion (lines 287-296). In short, our method relies on molecular and biophysical properties common to most lipids. Still, we acknowledge the effect of lipid composition as an issue for future studies.

3: We do focus on fully disordered IDRs, but have made an initial attempt at predictions for IDRs that contain amphipathic helices – the parameter values are changed in the latter case (line 134 vs. line 159). The web server has fully disordered IDRs as default but keeps the case of amphipathic helix as an option.

4: We now explicitly state the net-charge based grouping as the main limitation of our method (lines 279-280). Along with the two proteins in PMID 25846210, we now test our method against experimental data on six more proteins (lines 170-206; new Figure 2). Thank you for suggesting this example!

5: We have now expanded the opening paragraph to better explain how membrane interactions affect protein functions (lines 29-48), including more biochemical context, more mechanistic connection, and more examples. PMID 28601983 is now cited as Ref. 1.

6: We now cite PMID 16319880 (Ref. 3) as well as a number of other papers on IDR interactions with acidic membranes (lines 26-59). To distinguish our method from a toy model based on clustering of positively charged residues, we actually implemented such a model (lines 92-98 and new Figure S2). This toy model indeed predicts a correct trend, but our method increases accuracy by 2 to 3-fold (lines 140-141). No general trend could be found for clustering of aromatic residues.

7: Very good point! We now first present the toy charge model as a stepping stone, and then explain how our method is different (lines 101-107).

Round 2

Reviewer 1 Report

Accept to publish. 

Reviewer 2 Report

I was initially sceptical about this paper, but the authors have done a great work to address my concerns. Especially the benchmarking against NMR data in figure 2 adds a lot of confidence to the performance of the algorithm.